**Data Availability Statement:** All relevant data are within the manuscript and its Supporting Information files.

# Dog ecology and rabies knowledge, attitude and practice (KAP) in the Northern Communal Areas of Namibia

Tenzin Tenzin[1], Emmanuel H. Hikufe[2], Nehemia Hedimbi[3], Rauna Athingo[4], Mainelo Beatrice Shikongo[5], Thompson Shuro[6], Johannes Iipinge[7], Nelson Herman[7], Matias Naunyango[8], Frenada Haufiku[9], Josephat Peter[10], Laina Hango[10], Sara Gottlieb[11], Kenneth Shoombe[4], Nicolai Denzin[12], Frank Busch[13], Frederic Lohr[14], Moetapele Letshwenyo[1], Gregorio Torres[15], Conrad M. Freuling🔘[16]*, Thomas Müller[16], Albertina Shilongo[2]

**1** World Organisation for Animal Health (WOAH), Sub-Regional Representation for Southern Africa, Gaborone, Botswana, **2** Directorate of Veterinary Services, Ministry of Agriculture, Water and Land Reform, Windhoek, Namibia, **3** State Veterinary Office, Ministry of Agriculture, Water & Land Reform, Directorate of Veterinary Services, Kunene region, Opuwo, Namibia, **4** Animal Disease Control—North, State Veterinary Office, Ministry of Agriculture, Water & Land Reform, Directorate of Veterinary Services, Ongwediva, Namibia, **5** State Veterinary Office, Ministry of Agriculture, Water & Land Reform, Directorate of Veterinary Services, Zambezi region, Katima Mulilo, Namibia, **6** State Veterinary Office, Ministry of Agriculture, Water & Land Reform, Directorate of Veterinary Services, Kavango East region, Rundu, Namibia, **7** State Veterinary Office, Ministry of Agriculture, Water & Land Reform, Directorate of Veterinary Services, Oshana region, Ondangwa, Namibia, **8** State Veterinary Office, Ministry of Agriculture, Water & Land Reform, Directorate of Veterinary Services, Ohangwena region, Eenhana, Namibia, **9** State Veterinary Office, Ministry of Agriculture, Water & Land Reform, Directorate of Veterinary Services, Oshikoto region, Omuthiya, Namibia, **10** State Veterinary Office, Ministry of Agriculture, Water & Land Reform, Directorate of Veterinary Services, Omusati region, Outapi, Namibia, **11** State Veterinary Office, Ministry of Agriculture, Water & Land Reform, Directorate of Veterinary Services, Kavango East region, Nkurunkuru, Namibia, **12** Institute of Epidemiology, Friedrich-Loeffler-Institut (FLI), Greifswald-Insel Riems, Germany, **13** Institute of International Animal Health/One Health, Friedrich-Loeffler-Institut (FLI), Greifswald-Insel Riems, Germany, **14** Mission Rabies, Cranborne, United Kingdom, **15** World Organisation for Animal Health (WOAH), Paris, France, **16** Institute of Molecular Virology and Cell Biology, Friedrich-Loeffler-Institut (FLI), WHO Collaborating Centre for Rabies Surveillance and Research, WOAH Reference Laboratory for Rabies, Greifswald-Insel Riems, Germany

* Conrad.Freuling@fli.de

## Abstract

In 2021, a comprehensive dog demographic questionnaire combined with a KAP survey were conducted in the northern communal areas (NCAs) of Namibia with the aim of gaining a better understanding of dog populations, owner behaviour, and knowledge, attitudes and practices (KAP) relating to rabies. The survey of 3,726 households across the eight regions of the NCAs provided insights that will inform interventions in order to improve human rabies prevention and Namibia's dog rabies control strategy. The results showed a relatively low average human/dog ratio (HDR) of 5.4:1 indicating a surprisingly high dog population of at least 272,000 dogs in the NCAs, 93% of which appear to be owned but are free-roaming. Data analysis revealed opportunities but also highlighted needs for improvements in rabies surveillance and mass dog vaccinations. Although knowledge, attitude, and practice scores towards epidemiologic and clinical aspects, human rabies prevention, and dog rabies vaccination were deemed to be acceptable, the survey nevertheless revealed deficiencies in certain aspects in some of the population. Interestingly, data seemed to indicate relatively high

**Funding:** This research was funded by the German Federal Ministry of Food and Agriculture German to the World Organization for Animal Health (WOAH) to (grant RIE-0712) to TT, CMF and TM. The funders had no role in study design, data collection and analysis, decision to publish, or preparation of the manuscript.

**Competing interests:** The authors have declared that no competing interests exist.

dog bite incidences per 100,000 people, ranging between 262 and 1,369 and a certain number of unreported human rabies cases. Despite the very high number of dogs, only 50% of dog-owning households reported having vaccinated their dogs. In order to address these issues, the planning, announcement, and implementation of mass dog vaccination campaigns needs to be adapted to achieve adequate vaccination coverage. Another focus needs to be on rabies awareness and education if Namibia is to be significantly contributing to the global goal of "Zero by 30".

## Author summary

As a neglected disease, rabies remains a major problem in Africa and Asia. Here we report the results of an extensive community survey on dog ownership and knowledge, attitudes and practices (KAP) related to rabies control and prevention, covering the Northern Communal Areas (NCAs) of Namibia. The survey conducted in 2021 included more than 3,700 households and provided useful insights that will inform interventions in order to improve human rabies prevention and Namibia's dog rabies control strategy. Our results show that there was 1 dog for about every 5 humans, of which the vast majority is free-roaming. This surprisingly high dog population is not only supporting disease transmission but further complicating control efforts. Most people were aware that dog-mediated rabies is present in this part of Namibia and had an acceptable attitude and behavior towards it. Depending on the region, between 262 and 1,369 people per 100,000 inhabitants were bitten by dogs during the survey period. Although the number of dog bites is relatively high, more than 90% of victims stated that they had sought hospital treatment after being bitten. However, there are still gaps in laboratory-based surveillance of dog-mediated rabies, and incomplete or lack of rabies prophylaxis after dog bite injuries have led to unreported human deaths. Therefore, improved vaccination measures for dogs, consistent rabies prophylaxis after dog bite injuries as well as awareness-raising measures to increase people's knowledge and awareness are necessary if human deaths caused by rabies are to be permanently prevented.

## 1. Introduction

Dog-mediated rabies has long been a major socioeconomic and public health threat for people in low- and middle-income countries of Africa and Asia. Although it is an entirely vaccine-preventable disease, tens of thousands of people still die each year from rabies, which is usually transmitted to humans through bites from domestic dogs in these regions [1,2]. However, estimating the true burden of dog-mediated rabies in humans is difficult because of drastic under reporting due to inadequate surveillance in most countries where dog-mediated rabies is endemic [3]. Recognizing that rabies in dogs can be controlled with available resources, the international community, led by the Tripartite [World Health Organization (WHO), World Organization for Animal Health (WOAH) and Food and Agricultural Organization of the United Nations (FAO)], has agreed on a global strategic plan, in line with the United Nations Sustainable Development Goals, to end dog-mediated rabies in humans by 2030 [4,5].

In Namibia, dog-mediated rabies is endemic and mainly confined to the Northern Communal Areas (NCAs), where it has caused more than two hundred human rabies deaths since the beginning of the millennium [6]. To address this increasingly problematic situation, the

Namibian government implemented a dog rabies control program in the NCAs in 2016 [7,8]. While the pilot project in the Oshana region and the initial roll-out phase were considered a great success, progress in controlling dog-mediated rabies has stagnated in recent years [9]. This is partly because of the SARS-Cov-2 pandemic, as well as recurrent outbreaks of foot-and-mouth disease (FMD) and contagious bovine pleuropneumonia (CBPP) in parts of the NCAs. This required concerted actions by both public and animal health in an attempt to bring the situation under control [10,11]. As a result, parenteral mass dog rabies vaccinations (MDV) planned for the years 2020–2022 were jeopardized as resources had to be diverted [9]. However, even in the few areas in the NCAs where MDV campaigns could be conducted after all, follow-up studies showed that vaccination coverage rates in dogs were below the thresholds needed for rabies control and elimination [9], indicating inadequacies of this approach in resource-poor settings [12]. The reasons and challenges can be many, ranging from infrastructural issues due to the geographic location (dispersed) of the region, to the level of awareness in the population and knowledge of the density of susceptible dog populations, to maintaining adequate herd immunity in free-ranging dog populations, to name a few [12–14].

Knowledge, Attitudes and Practices (KAP) surveys are a quantitative method (predefined questions formatted in standardized questionnaires) that are widely used to gather quantitative and qualitative information for effective planning of public and animal health intervention programs [15–17]. Objectives may include provision of baseline data for planning, implementing and evaluating national control programs, identifying knowledge gaps, cultural beliefs, and behavior patterns and barriers to infectious disease control, and designing public health or disease awareness campaigns [18]. Numerous KAP surveys on rabies have been published from African countries with widely varying targets, e.g., Benin [19], Burkina Faso [20], Cameroon [21], Chad [22], Côte d'Ivoire [23], Democratic Republic of the Congo [24], Ethiopia [25–28] Ghana [29], Mali [30], Morocco [31], Nigeria [32], Rwanda [33], Senegal [34] Tanzania [18,35], Uganda [36,37], and Zimbabwe [38]. In Namibia, rabies tailored KAP surveys have been conducted only on a small scale, e.g., individual towns or constituencies in the NCAs [39,40].

We hypothesized that knowledge about baseline data and owners' attitudes towards dog vaccination and post-exposure prophylaxis (PEP) translates into improved, optimized and refined rabies control and prevention strategies. Thus, the first objective of this large-scale cross-sectional community survey was to gain a better understanding of the dog demography and the human/dog ratio (HDR) in the affected areas to provide more realistic estimates of dog population sizes in different settings using nationally available human census data. The second objective was to obtain up-to-date information on community members' knowledge, attitudes, and practices regarding rabies under the conditions of the national dog rabies control program implemented in the NCAs. Specifically, we wanted to determine if there were any knowledge gaps, misconceptions, or misunderstandings that might hinder current rabies control implementation, acceptance, and behavior change. Another focus was to receive information about dog bite incidents and associated post-exposure practices of community members as a basis for implementing an integrated bite case management (IBCM) pilot project.

## 2. Materials and methods

### 2.1. Ethical and legal considerations

Research permission and ethical clearance was obtained from the Directorate of Veterinary Services (DVS) within the Ministry of Agriculture, Water, and Land Reform Namibia (MAWLR) (CVO 14 April 2021) and from the National Commission on Research Science and Technology, Namibia (file reference AN202101020). The study followed established

procedures in Namibia related to statistical surveys (Statistics Act 9 of 2011) [41]: As no sensitive individual information or clinical samples were collected from participants, the requirement for signed, informed consent was waived by the National Commission on Research Science and Technology. Permissions to visit the respective communities was granted from both official local and traditional authorities prior to the initiation of the research at the respective constituencies. Prior to the individual interview, respondents were informed of the objectives of the study, advised that participation was voluntary, and that all data collected would be kept confidential. Subsequently, oral consent was obtained and documented in the mobile phone app. Only participants over 18 years of age were interviewed. If the selected respondent did not orally consent to be interviewed, the next respondent was selected and interviewed.

## 2.2. Study design and setting

A cross-sectional study was conducted from April to June 2021 in the eight regions of NCAs, i.e., Kunene, Omusati, Oshana, Ohangwena, Oshikoto, Kavango West, Kavango East, and Zambesi, which are sub-divided into 75 constituencies. The study area covered approximately 263,376 km$^2$ and included the entire implementation area of the national dog rabies elimination program representing 31.9% of Namibia's territory [8]. According to the 2016 Namibia population and housing census, these regions are inhabited by about 1.32 million people representing 56.9% of the country's population with an average household (HH) size of 4.48 persons. The average population density in the NCAs ranges between 0.85 people/km$^2$ (Kunene) and 23.87 people/km$^2$ (Ohangwena) [42].

In order to estimate the proportion of the assessed parameters among the overall HH in the study area from a sample of HH, the sample size was calculated based on the worst-case assumption of a 50% prevalence of parameter occurrence. Conservatively, the number of households was assumed to be infinite, and an accepted error of 5% with 95% confidence was chosen. Based on established calculation methods [43] these specifications yielded a sample size of 385 per region of the NCA, which resulted in a total sample size of 3,080 HHs to be surveyed. In the absence of clearly defined administrative boundaries for villages/settlements and an official HH register for NCAs, so-called "crush pens" (n = 194) were used as a starting point for selecting HHs to be interviewed and randomly selected using available GIS layers. Crush pens are uniformly distributed, permanent facilities within NCAs that are regularly used as vaccination sites for cattle but also for targeted mass vaccination campaigns for dogs [8,10,44]. To achieve the sample size, survey teams were required to interview at least 15 HH in the vicinity of selected crush pens. To cope with assumed potentially incomplete or compromised data sets, the sample size was increased to 20 HH per crush pen.

## 2.3. Data collection

Data were collected using a self-designed, multi-structured questionnaire to assess dog demographics and KAP towards rabies. Closed multiple-choice questions and variables were selected to capture details on individual and HH characteristics in order to assess socioeconomic status and education level [18,45,46]. The questionnaire consisted of five sections. Here, specific questions addressed respondents' sociodemographic background and characteristics (Section 1); dog demographics, i.e., dogs living in the HH at the time, including information on dog ownership, management, and vaccination (Section 2); and respondents' knowledge, attitudes, and practices regarding rabies, rabies prevention strategies, actions toward animals suspected of being rabid, and incidents of animal bites in the HH in the past two years (Sections 3, 4, and 5). The questionnaire is available in the Supplementary Materials (S1 Table).

Door-to-door surveys were conducted by a total of 37 enumerators (between 3 and 6 per region) in teams of two over a period of 5 to 7 days in each region. Since the settlement areas were widely scattered, the direction taken by the survey teams from the crush pen was randomized. Along the route, every third house was surveyed, taking predefined turns at successive road junctions until the required number of HHs to be surveyed in the area was reached. One adult member per HH was selected for the survey, which was conducted in the local language or in English, depending on the preference of each respondent. Respondents had the option to stop the interview at any time despite their initial consent.

Survey data were collected via mobile phones using the Worldwide Veterinary Service smartphone App (WVS Data collection App) essentially as described [44,47]. The App and its template were kindly provided by the non-governmental organization Mission Rabies (https://missionrabies.com/). The questionnaire form was pre-designed by an administrator on the backend platform and integrated into the WVS Data collection App and remotely loaded to the handsets using 3G. Data were entered offline and synchronized via WiFi or mobile data to a web-based server once an internet connection was available and uploaded online. Enumerators were trained in mobile App supported survey techniques during a pilot study. The survey coordinator monitored survey progress in the field by accessing the App database and provided technical advice and feedback to survey teams via WhatsApp chats and calls [47].

## 2.4. Data analysis and statistics

Data collected during the study were downloaded from WVS software in comma-separated value files (CSV), checked for errors and analyzed using R software (version 3.50). Logistic regression analyses were performed to assess the factors associated with dog ownership status, dog vaccination status, having heard of rabies, dog bite and dog meat consumption practices as a binary outcome using the socio-demographic characteristics as explanatory variables (sex, age, qualification, occupation, livestock ownership, region, settlement type/residence). First, a univariable logistic regression analysis was performed and explanatory variables with p-values ≤0.20 were included in a multivariable model. The variables with p-value < = 0.05 were considered significant. Model fitting was conducted using the Hosmer-Lemeshow test with estimates presented as adjusted odd ratios (AOR) with corresponding 95% confidence interval (CI).

Using the number of dogs and people identified per HH surveyed, the proportion of dog-owning HHs (DOHHs), the HDR and the dog:HH ratio in the study area were estimated [48,49] with confidence limits calculated for proportions and means [50,51]. Population and HH data from the 2016 Inter Censal Demographic Survey (NIIDS) [42] were used to estimate the total dog population in the NCA by region and constituency according to recent literature [52]. The dog population was also projected using the annual growth rate for 2021 according to the Namibian Statistics Agency. The human population density at the location of the individual survey was derived from the Gridded Population of the World, Version 4 (GPWv4) [53].

Respondents of the KAP survey were categorized into two groups, below and above average knowledge, for each of the three areas knowledge of, attitude towards and practice with rabies. Knowledge and attitude were assessed only for those who were generally aware of rabies as a disease, practice only for those who had experienced a dog bite themselves or in the HH. Categorization in the aforementioned areas was based on six, two and seven multiple choice questions, respectively. Questions were scored through an evaluation of each answer. Points, i.e. one point for each correct and zero for each incorrect answer, were awarded for each selected correct answer as well as each non-selected incorrect answer with a maximum of 23, 12 and 17

achievable points, respectively. The mean of the scored points was calculated for each area and used as a threshold for performance dichotomization. Scores below average were labelled negative or poor [24,54]. Descriptive statistics were calculated as frequency, percentage, point estimates, mean and inter quartile ranges (IQR). Data were analysed using Pearson chi-square test, Pearson correlation, independent T-test, and one-way analysis of variance (ANOVA) test. At 95% Confidence Interval, a p value < 0.05 was considered to be statistically significant.

# 3. Results

## 3.1. Sociodemographic characteristics of respondents

The dog demography and KAP survey was delivered to 3,771 HHs in the NCAs, of which 3,726 HHs (98.8%) consented to be interviewed representing a population of 29,892 people. The survey covered 55 of 75 constituencies of the NCAs (Fig 1).

The gender ratio was completely balanced with 50% women, equaling one man to one woman. Respondents represented all age groups most of whom were farmers (42.08%) or unemployed (24.37%) and reported having primary (26.14%) or secondary education (42.08%) (S2 Table and Fig 2). The average size of families (people living in one HH) was 8.02 people (range 1–70) with an average of 3.9 children (under 18 years of age) per family.

## 3.2. Dog demography

Two-thirds (66.5%; 2477/3726, CI 64.9–68.0) of HHs surveyed in the NCAs reported keeping dogs (n = 5483). The proportion of DOHHs in rural HH (2059/2972, 63.3%, CI 67.6–70.9) was

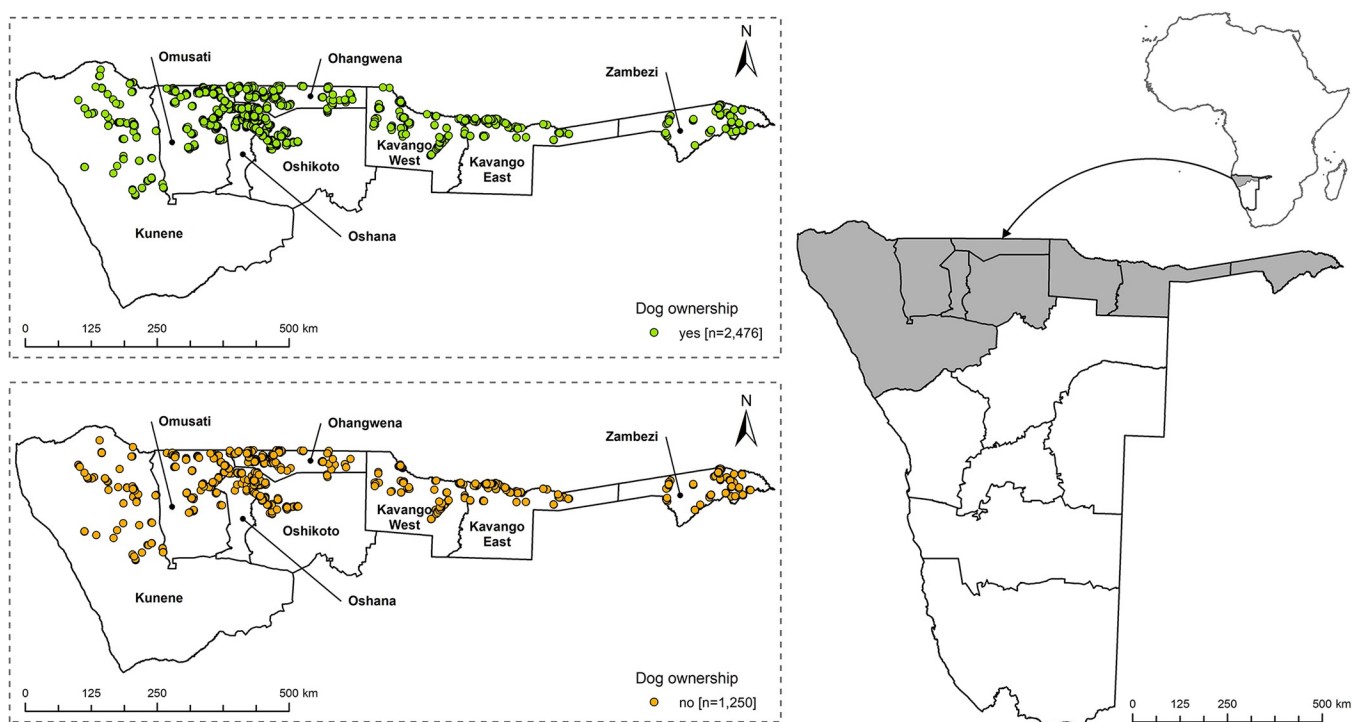

**Fig 1.** Map of Africa and Namibia showing the eight regions of the Northern Communal Areas (NCAs, right). In the zoom outs, the location of the surveyed households (HH) in the NCAs from April to June 2021 is shown. HHs without dogs and dog owning households (DOHHs) are highlighted in green and orange, respectively (right). Map content was produced with Esri ArcGIS software using study data and data provided by GADM available online: https://gadm.org/download_country.html.

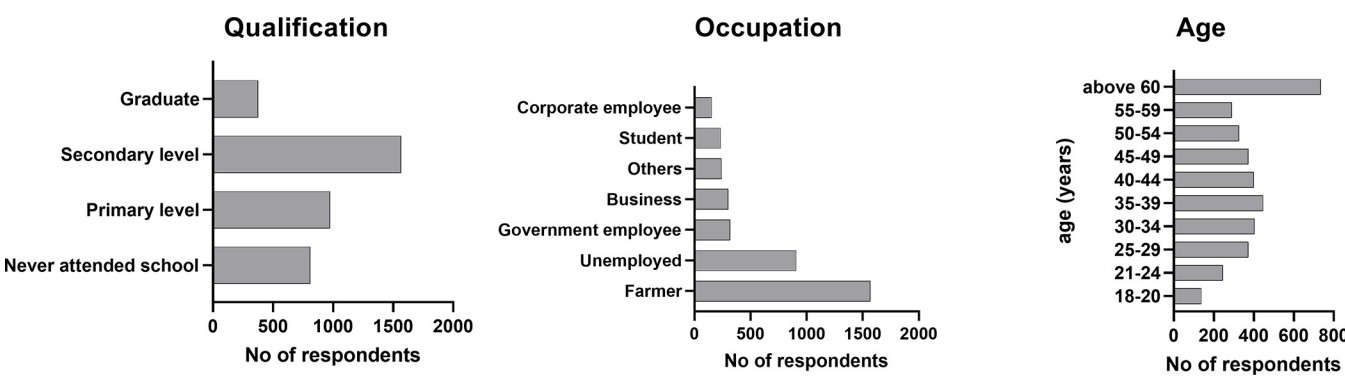

**Fig 2. Socio demographic characteristics of 3726 HH's respondents in NCA, Namibia, 2021.**

significantly higher [X2 (1, N = 3726) = 52.7, p < .001] than in urban HH (417/754, 55.3%, CI 51.7–58.9) (S3 Table). However, there was no spatial clustering of DOHHs and non-DOHHs across the regions, but a complete overlap (Fig 2). In terms of age structure, 77.8% (4265/5483) of the dogs were adult (>1 year) and 22.2% (1218/5483) sub-adult/puppies (< 1 year). The male:female ratio was 1.54:1. The vast majority of respondents (90.5%) reported that they keep dogs to guard properties and let their dogs roam free (90.3%), while only a few confine their dogs day and night.

The average number of dogs per DOHH was 2.21 (5483/2476, CI 2.14–2.28) with a range of 1 to 19. This resulted in an overall dog:HH ratio of 1.47 (5483/3726, CI 1.41–1.53) and a HDR of 5.45 (29,892 people/5483 dogs, CI 5.37–5.54) for the NCAs (range 4.48–6.96 at a regional level) (Fig 3). When the HDR was calculated per HH surveyed and correlated to the human population density (people/km$^2$), there were no differences between rural and urban areas (Fig 4).

Using the estimated HDR of 5.45 from the survey and assuming a total human population of 1,324,290 according to the 2016 Census or the projected human population data for 2021 with an annual growth rate of 1.86% (according to the 2019 Census, n = 1,480,204), the total number of dogs in the NCAs was estimated to range between 242,875 (95% CI: 239,167–246,583) and 271,597 (95% CI 267,325–275,614), respectively, with constituencies in the

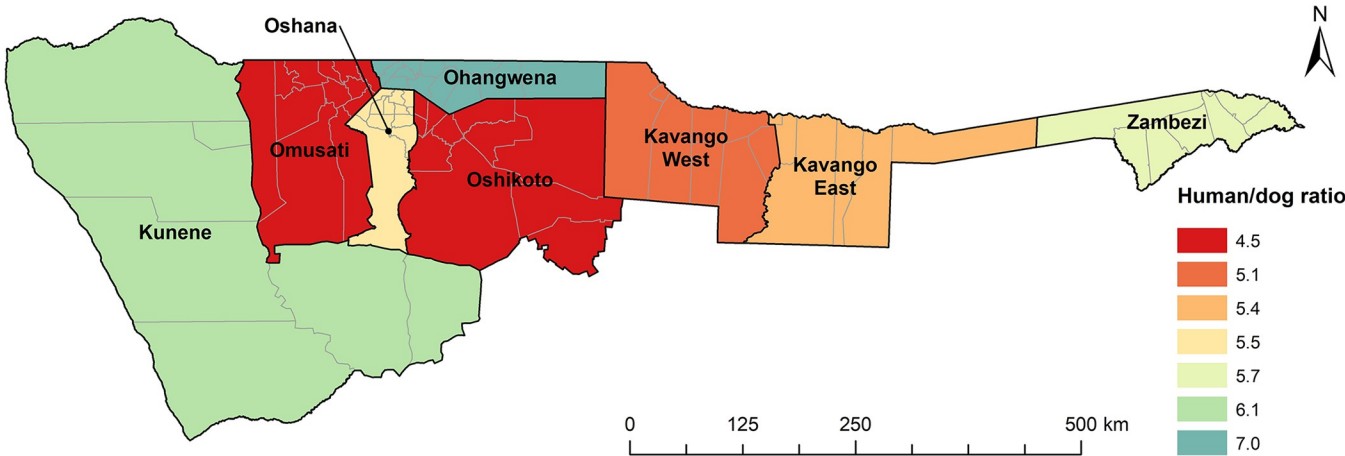

**Fig 3. Calculated HDRs for the NCAs according to regions as per survey data.** Borders of constituencies are indicated. Map content was produced with Esri ArcGIS software using study data and data provided by GADM available online: https://gadm.org/download_country.html.

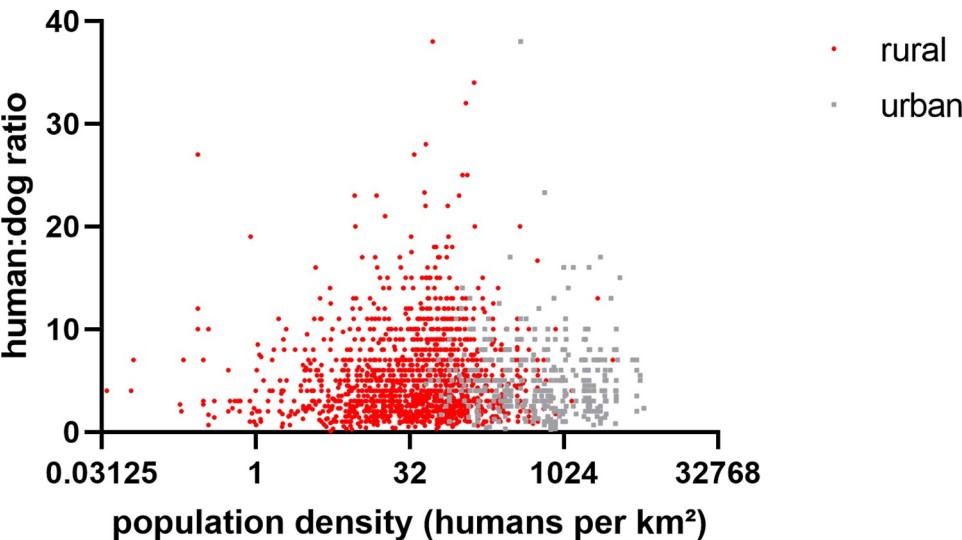

**Fig 4. Comparison of HDR per HH differentiated between rural and urban in the survey.** The human population density at the location of the individual survey was derived from the Gridded Population of the World, Version 4 (GPWv4) [53].

Omusati, Ohangwena und Oshikoto region having the highest numbers of dogs (S1 Fig). This resulted in an estimated overall dog density of 0.94 dogs/km$^2$ for the entire NCAs, with dog densities being highest in the urban areas of Katima Mulilo (139/km$^2$), Rundu (89/km$^2$), Ondangwa (85/km$^2$), Ongwediwa (34/km$^2$), and Oshikato (33/km$^2$) (S2 Fig).

If the average dog:HH ratio of 1.47 according to the survey and the number of HHs as per 2016 (n = 294,698) and 2019 (n = 279,280) census is used for calculation, the dog population for the NCAs would range between 410,541 (95% CI 393,785–427,298) and 433,206 (95% CI: 415,524–450,888) resulting in a dog density of about 1.65 dogs/km$^2$.

### 3.3. Dog rabies vaccination

Survey records indicate that 49.6% (1228/2476) of DOHHs reported that their dogs were vaccinated during the 2020 mass dog vaccination campaign, representing an overall vaccination coverage rate of 38.6% (2118/5483) based on the number of dogs identified (S2 Table). A multivariable logistic regression model showed that dog-owning respondents from urban settlements were more likely to have their dogs vaccinated (OR = 2.7; 95% CI = 2.1–3.6; P<0.001) than respondents from rural settlements. Significant associations with dog vaccination were also found among persons owning livestock (OR = 2.1; 95% CI = 1.7–2.7; P<0.001), male respondents (OR = 1.6; 95% CI = 1.2–2.8; P<0.001), and persons who had "heard of rabies" (OR = 2.1; 95% CI = 1.7–2.7; P = 0.017) (S4 Table). The fixed-effects multivariable model appeared to fit the data adequately (Hosmer-Lemeshow goodness-of-fit test statistic (GOF) = 7.626, degree of freedom (DF) = 8, P = 0.471). However, there were regional differences; vaccination coverage was higher in urban areas (52.4%, 444/847) than in rural areas (36.1%, 1674/4636), with the fewest dogs vaccinated in the Zambezi region (18.9%, 92/486) and the most dogs vaccinated in the Oshana region (57.1%, 368/645) (S4 Table).

When asked about the approximate distance between their residence and dog vaccination sites, 69.1% (848/1228), 24.3% (299/1228), and 6.3% (77/1228) of respondents, respectively, indicated that it was < 1 km, between 2 and 3 km, and 4 km or more, respectively. About 60.1% (738/1228) and 55.8% (685/1228) of dog owners cited the radio and veterinary

personnel, respectively, as sources of information about vaccinations. In contrast, of the DOHHs who did not vaccinate their dogs during the 2020 vaccination campaign, 43.6% (544/1249) reported that they were not aware of MDV campaigns. When asked about causes of death, 7.7% (67/870) of respondents from DOHH having affirmed dog fatalities in 2020 indicated that their dogs most likely died from rabies.

### 3.4. Knowledge, attitude and practices survey

While the majority (87.3%, 3252/3726) of respondents in all regions had heard of rabies, the level of knowledge about rabies among respondents was heterogeneous. With a score below the mean of 12.6 points (range 3 to 23; IQR 10–14), 53.1% of respondents were moderately to poorly informed, while 46.9% were classified as well informed (Fig 5). Good knowledge about rabies was strongly associated with gender (male; OR = 1.5; 95% CI = 1.3–1.7; P<0.001), residence setting (urban; OR = 1.7; 95% CI = 1.4–2.0; P<0.001) and dog ownership (OR = 1.2; 95% CI = 1.1–1.4; P<0.016), while poor knowledge is often linked to limited schooling (OR = 0.7; 95% CI = 0.6–0.9; P = 0.0007) (GOF = 10.635, DF = 8, P- value = 0.262) (S6 Table).

Responses to the attitude statements indicated that most people had a correct attitude toward encountering rabid dogs or being bitten by rabid dogs. The obtained minimum score was 3 out of 12 points with a mean score of 6.3 (median = 6) 1st IQR = 5, 3rd IQR = 7 (Fig 6).

Using the mean score as the cut-off 36.5% had a favorable attitude towards rabies. Residence (urban areas; OR = 1.8; 95% CI = 1.5–2.5; P<0.001) and a good knowledge score about rabies (OR = 2.5; 95% CI = 2.2–2.9; P<0.001) had a significant positive effect on 'attitude towards rabies', while poor attitude were linked to rudimentary education (OR = 0.7; 95% CI = 0.6–0.8; P = 0.00038) (GOF = 5.271, DF = 8, P- value = 0.728) (S7 Table).

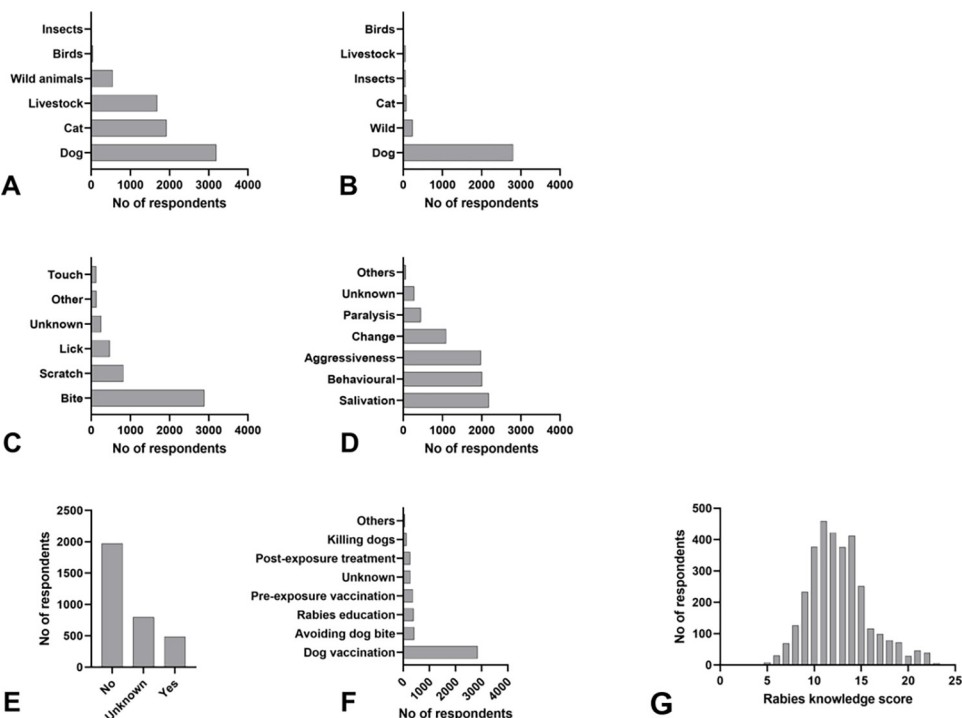

**Fig 5. Respondents' responses to the question about rabies.** A: Which animals can get rabies?; B: What is the main source/vector of rabies?; C: What is the mode of transmission?; D: What are the clinical signs of rabies?; E: Can rabies be treated?; F: What prevention methods do you think are the most appropriate?; G: Distribution of respondents total knowledge score.

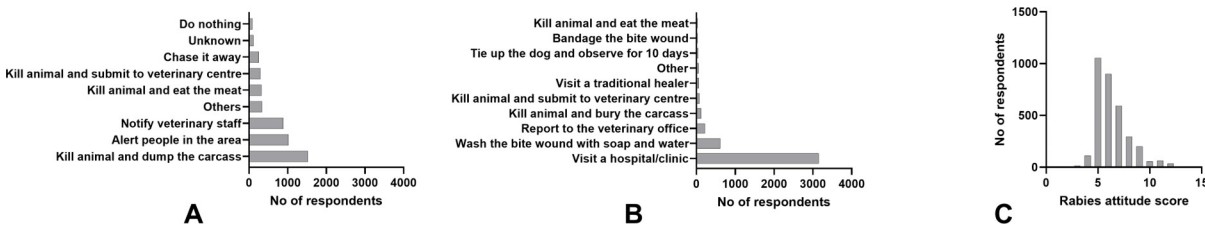

**Fig 6.** Answers of respondents when ask what they would do if they (A) encountered a suspect rabid dog and (B) were bitten by a dog. Distribution of respondents' total attitude score (C).

### 3.5 Dog bite incidence and response

A total of 403 respondents (10.8%) reported dog bites in the past two years, resulting in an overall annual dog bite rate of 674 (95% CI 612–743) per 100,000 residents, with incidences above average reported from the Kavango East and Zambezi regions (S5 Table and Fig 7).

**Fig 7. Graph depicting the dog bite incidence in the NCAs of Namibia.** Mean (circle) and 95% CI (whiskers) are indicated. The dashed line and grey area symbolize the mean and 95% CI for the total NCAs.

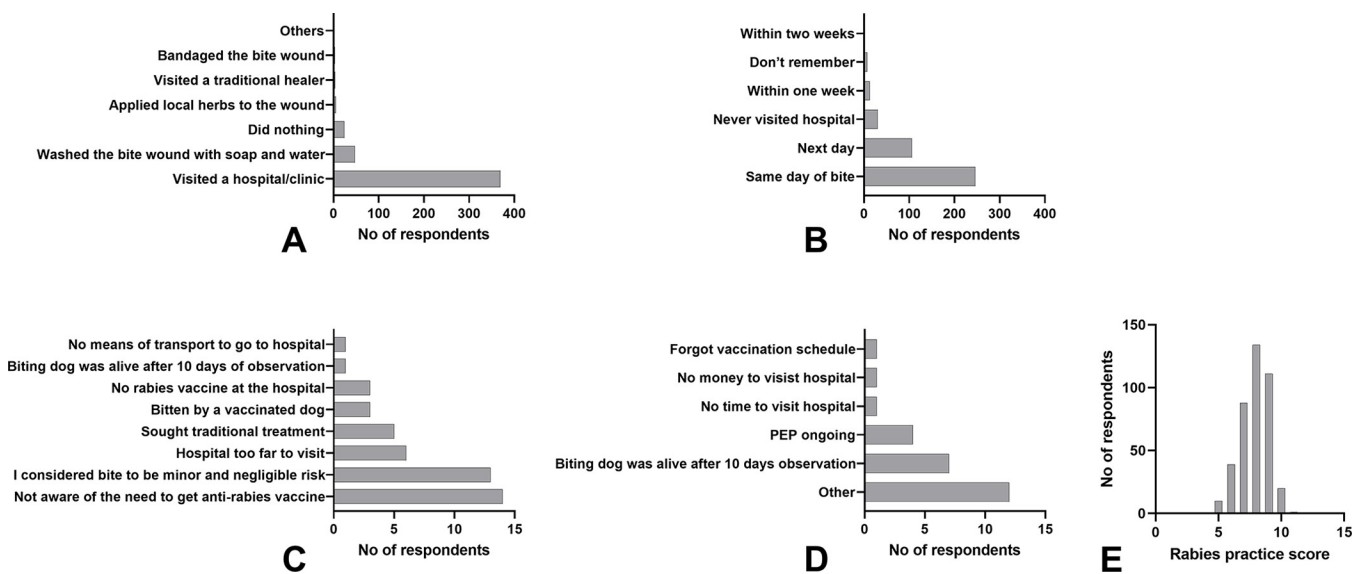

**Fig 8.** Respondents' practice behaviors related to immediate action/measures after a dog bite (n = 403) (A), time elapsed until hospital visit (n = 369) (B), reasons for not seeking medical help at a hospital (n = 43) (C) reasons for not completing a PEP course (n = 29) (D). Distribution of respondents total practice score (E).

Dog bite incidents were reported for all HH members, and included provoked bites (40.4%; 163/403), unprovoked bites (53.6%, 216/403), while 5.9% (24/403) of respondents could not recall the cause of the bite incidents. Men were less likely to experience and/or report dog bites (OR: 0.8; 95% CI: 0.6–0.9; P-value = 0.015) than women (GOF = 0.0086, DF = 8, P-value = 1.000). In 21.8% of cases, the biting dog was proven to be vaccinated, in 36.2% was unvaccinated, and in 41.9% the vaccination status of the biting dogs was unknown. In one third of the cases the biting dogs were subsequently killed. When asked, only 4.3% of respondents indicated that they had submitted these dogs for laboratory testing.

Responses related to behavior patterns, i.e. bite wound management and health seeking behavior in the case of bite exposure, are depicted in Fig 8. Respondents scored a minimum of 5 and a maximum of 11 points, with a mean of 7.9 (median = 8) IQR: 7–9. Thus, 66% were identified to have 'good practices', while 34% (137/403) were rated with poor practices related to dog bites and management. Although the majority of respondents (91.6%, 369/403) reported having visited a hospital after a bite incident, 12.2% did not complete the follow-up. A total of 46 respondents did not seek medical attention as they were unaware of the need for post-exposure rabies vaccination (30.4%) or considered the bite to be minor and of negligible risk (28.3%). Respondents reported that a total of seven bite victims in Kavango East (1), Ohangwena (3), Oshana (1), and Zambezi (2) died from animals suspected of having had rabies during the past two years in their communities.

## 4. Discussion

By combining the dog demographic questionnaire and the KAP survey, a detailed picture of dogs living in HHs and communities of the NCAs and their owners' knowledge, attitudes, and practices regarding rabies was obtained. This differed from earlier studies in parts of the same region, where the focus of these surveys had been primarily on rabies knowledge and awareness, while issues important for rabies control and prevention, such as dog demographics and human behavior in case of bite exposure, had not been adequately addressed [39,40]. With

3726 HHs interviewed representing 2.2% of the local population living in the NCAs, this is one of the largest dog demography and KAP studies in the Southern African Development Community (SADC). Only in Tanzania the number of HHs interviewed (n = 5141) was higher [18]. Considering the size of the study area (Fig 1), the organization of this combined survey was logistically complex and required a considerable amount of time and effort from the 18 teams in addition to their day-to-day activities. Therefore, it is all the more remarkable that the survey could be conducted in a relatively short period of two months, considering the distances the teams had to travel for this large-scale door-to-door survey (Fig 2). Also, data collection via the WVS data collection app was critical, as it formed the basis for the automated, computerized assessment of the survey which could become a standard for similar future projects [44,47].

Information on dog demographics and dynamics are critical for developing and planning effective vaccination strategies that are tailored to the target dog population, in particular free-roaming dogs [55–57]. In order to improve the effectiveness of the implementation of the rabies control and prevention measures, it was incumbent to adequately assess the dog population in the NCAs; despite a national rabies control strategy already being in place [7,8] the dog population in Namibia could, unfortunately, never be accurately determined due to a lack of reliable data [8]. Based on this survey, the calculated HDR of 5.45 for the NCAs (range 4.48–6.96, Fig 3) reflects ratios reported for Guatemala [58], Chile [59], Zimbabwe [60], Madagascar [61], and Thailand [62]. The HDR is much lower as compared to previous estimates that ranged from 9.95 (rural) to 15.2 (urban) [63], but had recently been corrected to 8.3 (entire NCAs) [8]. The latter dog population estimates were derived from either mean ratio estimates for the region used to extrapolate population sizes or, if ratios were not available for a region, mean ratios from neighboring regions and countries were used to extrapolate population sizes [63]. Using the HDR derived from this survey and nationally available human census data, the total number of dogs in the NCAs lies within a in range of 247,000 to 272,000, which is much higher than previously estimated. Generally, the HDR does not vary much across the NCAs (Fig 3). Although the number of DOHHs is higher in rural areas, if exact human population data from the 2016 census are used, there is no difference in the HDR between the two main types of settlement structures at a HH level (Fig 4). This is in contrast to other studies and assumptions that suggest that HDRs are generally higher in urban areas than in rural settings [1,63,64] and may be unique to Namibia based on its settlement structure [65]. These data suggest that there are little sociocultural differences in dog ownership in the NCAs, despite the relatively large number of ethnic groups living in Namibia.

It is surprising that extrapolating the size of the dog population from the calculated dog/HH ratio and the number of HHs according to the 2016 and 2019 census data results in such a high discrepancy (factor of 1.6) compared to using the HDR. One explanation could be that there are major differences between the definitions of a HH in our KAP study as compared to national censuses or UN definitions as discussed [66]. In our study, a HH was defined as a group of people who normally live and eat together under one roof. Interestingly, however, the census report does not include definitions, which may explain the discrepancy between 294,698 and 279,280 HH for 2016 [42]. Thus, there is good reason to believe that the size of the dog population of about 433,000 dogs based on the dog/HH ratio is an overestimate for the NCAs and generally suggests that population estimates based on HDRs may be more reliable.

Indeed, unreliable data or underestimated dog populations make it difficult to plan MDV campaigns and estimate the resulting vaccination coverage. The new dog population size estimates calculated in this study likely explain the relatively low overall vaccination coverage observed in this study (38.6%). It is difficult to prove whether the survey information can be trusted, as the dog owners usually do not have the appropriate vaccination certificate and if

they do, it is often not clear whether this applied to the dogs living in the HH at the time of the interview. Using specially designed vaccination tracking devices, it was shown that vaccination coverage was sometimes even lower, which explains the stagnation in controlling dog-mediated rabies in recent years [9]. Similar problems occurred in Tanzania, where during 2013–2017, when vaccination coverage was monitored, only about 20% of vaccination sites achieved the recommended coverage rate of 70%, with an average coverage rate of about 50% [14]. Unsuccessful vaccination campaigns have also been reported from countries such as Chad, Kenya, Nigeria, and South Africa with vaccination coverages far below the optimum [67–70]. This highlights the challenge to increase and maintain herd immunity in dogs in these regions [12,14], especially considering that almost all dogs in this area are owned but free-roaming. Controlling dog-mediated rabies in the NCAs is extremely challenging due to the dispersed and more uniform distribution of settlements across the area and associated population structure [65]. Against this background it appears questionable that almost 50% of the DOHHs across all regions claimed their dogs to be vaccinated and almost 69% of the respondents stated the distance to vaccination points was <1 km.

Therefore, to increase vaccination coverage in dogs future MDV campaigns in NCAs will need to adjust the number of vaccine doses to match the dog population per constituency identified in this study, ideally with an upward safety margin, while reconsidering the number and strategic selection of locations for vaccination sites considering landscape and topography [71]. Monitoring MDV campaigns using specially designed vaccination devices for data collection and subsequent GIS analysis using gridded population data can help to better assess and, if necessary, improve vaccination coverage at the local level [9]. In view of the high proportion of hard-to-reach dogs and Namibia's extremely positive experience with the efficiency of oral vaccination of dogs from field trials [44,72], the long-term strategic integration of this vaccination variant into the national dog rabies control program should be seriously considered. Also, the fact that nearly half (43.6%) of DOHHs (n = 1228) were not aware of MDV campaigns and the other half did not vaccinate their dogs for various reasons raises questions about awareness and communication regarding these intervention measures. Regular community engagement and ongoing awareness of MDV are critical, and strategies must be adaptable and make the best use of all available resources [73] and help increase dog owner participation in vaccination campaigns, as recently demonstrated in Tanzania [74].

In comparison to the rather spread out human population and the size of the country Namibia has one of the best rabies surveillance systems in Africa [6,75]. With this in mind, it seems interesting to note that 7.7% of DOHH having affirmed dog fatalities in the previous year assumed that their dogs had most likely died from rabies. If projected to the entire HHs in the NCAs this would amount to more than 5,000 rabies suspect dogs. Even if only 50% of these were considered due to the uncertainty factor in defining HHs as mentioned above, it would still be quite a high number of supposedly suspected rabid dogs. However, rabies prevalence varies from region to region and in time, so extrapolation to the entire region may lead to an overestimation. Also, it remains unclear how many of these suspected rabid dogs were confused with diseases of similar neurologic signs. Regarding mortalities in dogs in reference to infectious diseases other than rabies, there is reason to believe that canine distemper virus (CDV), canine parvovirus (CPV), canine babesiosis, snakebites envenomation, and toxicosis which can also cause neurological signs similar to rabies, are present in the NCAs, as it is in other African countries [76,77]. However, it appears that the number of rabid dogs is somehow underestimated and surveillance including laboratory confirmation could still be improved.

In addition, it was important to see the extent to which previous interventions have changed public perceptions and attitudes regarding rabies control. The KAP survey revealed a rather heterogeneous picture: Given that the majority of respondents had scores below average

in the areas of knowledge (53.1%, Fig 5) and attitude (63.5%, Fig 6), the relatively positive performance of respondents (66%, Fig 8) in terms of practices is seemingly contradictory. One problem associated with this observation is that it may be biased because, unlike the knowledge and attitudes themes, which included all HHs (3726), only HHs with bite victims (407) were interviewed for the practices theme. One may argue that this may better reflect the actual situation, as respondents provided accurate information about what they specifically did when they were bitten by a dog. On the other hand, valuable information is lost when respondents are theoretically asked what they would do if bitten by a dog without prior experience.

The self-reported dog bite incidence per 100,000 people in the NCAs ranged between 262 in the Kunene Region and 1,369 in the Zambezi Region. Interestingly, there seems to be a west-to-east gradient with incidences much above average reported for the two easternmost regions (Fig 7). The reasons for this observation are elusive as no other factor assessed in this survey nor in a recent census [42] demonstrated such gradient. One plausible explanation could be that hunting with dogs and a more prevalent interface with wildlife modified the dogs' behavior resulting in more bite inflictions. Generally, the observed bite incidence is very high and comparable with other rabies endemic settings like in South Africa (400) [78], Bangladesh (628) [79] and Pakistan (935) [80]. Dog bite incidences reported from African countries e.g., Ghana (248) [81], Nigeria (200) [82], Kenya (248) [83] and Tanzania (60) [84] were lower, but only relied on hospital-based surveillance data. Also, it was observed that 1.7% of HHs with dog bites in the past 2 years reported a total of seven victims dying from rabies in some parts of the NCAs. If extrapolated, this would result in a human rabies incidence in humans of 19.8/100,000 inhabitants. This is in contrast to previous official reports which indicated a much lower rabies incidence in humans of 0–2.4/100,000 [6].

The majority of respondents (97%) declared they would seek medical advice, 92% visited a hospital after a bite incident, and 88% completed the full PEP course. This compliance is quite exceptional and in contrast to e.g., results from Uganda [37] where only 56% of the interviewees indicated that dog-bite victims should visit a hospital and only 3 percent received PEP. While traditional therapies may be an issue in other socio-cultural settings [37,85–87], in this study only six respondents (1.5%) declared that they sought traditional treatment, despite the fact that 8% (298/3726) mentioned that they were aware of various traditional methods of treatment relating to dog bites in humans but also regarding the treatment of dog bite wound in dogs (S8 Table).

Still, as any rabies victim is preventable, the respondents' practice patterns (Fig 8) and particularly their reasoning for not attending to a hospital clearly indicates that improvements in awareness and post exposure prophylaxis are needed. This requires closer cooperation between public health and veterinary services. Experience has shown that implementing an IBCM within a One Health framework can significantly improve rabies surveillance and performance and access to PEP in a region [88–93].

As a quantitative method, KAP surveys serve to gather information from representative segments of the population to uncover general behaviors including misconceptions or misunderstandings towards health activities implemented or to be implemented and associated behavioral changes. However, there are various challenges of conducting surveys in different settings. A major limitation is that KAP surveys essentially record respondents' opinions, which may not reflect the real scenario because people tend to provide answers that they think are right or that are generally accepted and appreciated with sensitive topics being particularly challenging [17]. Extrapolation to the general population should therefore be undertaken with caution. In addition, data is collected at a single point in time, i.e. 2021. Although two smaller KAP studies had already been conducted in the area, the focus was different [39,40] making it difficult to measure changes in the human population over time.

## 5. Conclusions

This large-scale survey offered valuable insights into dog populations sizes in the NCAs of Namibia as well as rabies related knowledge, attitude and practices of people living in this area. There are obvious deficiencies in all three of the latter topics, which need to be addressed by key stakeholders if rabies control and prevention is to be improved in the future. Targeted, large-scale awareness and education campaigns focused on information about the risks associated with dog-mediated rabies and the proper behaviors to avoid those risks could prevent unnecessary deaths. In a true One Health context, this requires a greater commitment by public health agencies as regards both prevention and post-exposure prophylaxis. Piloting of an integrated bite case management system should be considered. From a veterinary perspective, mass dog vaccination campaigns require more accurate planning based on realistic regional dog population sizes and more efficient approaches to achieve better vaccination coverage in dogs. Better strategic selection of vaccination sites, increased use of oral immunization in view of the large number of free-roaming dogs, but also increased involvement of external (national and international) partners in mass vaccination campaigns (outsourcing) in view of limited resources should be considered if substantial progress is to be made in the control of dog-mediated rabies in the near future. The results are obviously of great importance to Namibia, however, other communities in Africa can learn from these findings and some of the themes and suggestions for rabies control program improvements are certainly clearly transferable.

## Supporting information

**S1 Table. Questionnaire used in the study.**
(CSV)

**S2 Table. Sociodemographic characteristics of the participants (n = 3726).**
(DOCX)

**S3 Table. Number of dogs recorded during the survey and vaccinated against rabies in 2020 in different regions of the the NCAs.**
(DOCX)

**S4 Table. Univariable and final multivariable logistic regression model to determine the factors associated with dog vaccination among respondents of DOHHs (n = 2476).**
(DOCX)

**S5 Table. Dog bite reporting and resulting dog bite incidences for the regions of the NCAs.**
(DOCX)

**S6 Table. Univariable and final multivariable logistic regression model to determine the factors associated with the respondent knowledge of rabies (good vs fair to poor) in NCA, Namibia (2021).**
(DOCX)

**S7 Table. Final multivariable logistic regression model to determine the factors associated with the respondent attitude towards rabies (favourable vs unfavourable attitude) in NCA, Namibia (n = 3252).**
(DOCX)

**S8 Table. Respondents' statements regarding traditional treatment of dog bites and treatment of dogs bitten by other dogs (by region) in NCAs (n = 3726).**
(DOCX)

**S1 Fig. Map of the NCAs showing the estimated dog population per constituency based on the projected human population for 2021.** Map content was produced with Esri ArcGIS software using study data and data provided by GADM available online: https://gadm.org/download_country.html.
(TIF)

**S2 Fig. Map of the NCAs showing estimated dog density (dogs/km$^2$) per constituency.** Map content was produced with Esri ArcGIS software using study data and data provided by GADM available online: https://gadm.org/download_country.html.
(TIF)

## Acknowledgments

We would like to thank Lorenz Nake from World Organization for Animal Health (WOAH) headquarters for his continuous administrative support and help in all WOAH managed Namibian-German collaborations and joint research projects. The help of Patrick Wysocki in creating the maps is greatly acknowledged. The authors would also like to thank all enumerators involved in this large-scale KAP survey for their overwhelming willingness and efforts to make this unique study a success.

## Author Contributions

**Conceptualization:** Tenzin Tenzin, Rauna Athingo, Johannes Iipinge, Kenneth Shoombe, Frank Busch, Frederic Lohr, Conrad M. Freuling, Thomas Müller, Albertina Shilongo.

**Data curation:** Tenzin Tenzin, Emmanuel H. Hikufe, Nehemia Hedimbi, Nicolai Denzin.

**Formal analysis:** Tenzin Tenzin, Emmanuel H. Hikufe, Rauna Athingo, Nicolai Denzin, Frank Busch, Conrad M. Freuling, Thomas Müller.

**Funding acquisition:** Tenzin Tenzin, Moetapele Letshwenyo, Gregorio Torres, Conrad M. Freuling, Thomas Müller.

**Investigation:** Tenzin Tenzin, Nehemia Hedimbi, Rauna Athingo, Mainelo Beatrice Shikongo, Thompson Shuro, Johannes Iipinge, Nelson Herman, Matias Naunyango, Frenada Haufiku, Josephat Peter, Laina Hango, Sara Gottlieb, Thomas Müller.

**Methodology:** Tenzin Tenzin.

**Project administration:** Tenzin Tenzin, Nehemia Hedimbi, Rauna Athingo, Kenneth Shoombe, Moetapele Letshwenyo, Gregorio Torres, Albertina Shilongo.

**Resources:** Kenneth Shoombe, Frederic Lohr, Albertina Shilongo.

**Software:** Frederic Lohr.

**Supervision:** Tenzin Tenzin, Kenneth Shoombe, Moetapele Letshwenyo, Gregorio Torres, Albertina Shilongo.

**Validation:** Tenzin Tenzin, Nicolai Denzin, Conrad M. Freuling.

**Visualization:** Tenzin Tenzin, Conrad M. Freuling.

**Writing – original draft:** Tenzin Tenzin, Frank Busch, Conrad M. Freuling, Thomas Müller.

**Writing – review & editing:** Tenzin Tenzin, Emmanuel H. Hikufe, Mainelo Beatrice Shikongo, Thompson Shuro, Johannes Iipinge, Nicolai Denzin, Frank Busch, Conrad M. Freuling, Thomas Müller, Albertina Shilongo.

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
