## [Decision Letter · Decision Letter 0]

27 Dec 2023

Dear Dr. Freuling,

Thank you very much for submitting your manuscript "Dog Ecology and Rabies Knowledge, Attitude and Practice (KAP) in the Northern Communal Areas of Namibia" for consideration at PLOS Neglected Tropical Diseases. As with all papers reviewed by the journal, your manuscript was reviewed by members of the editorial board and by several independent reviewers. The reviewers appreciated the attention to an important topic. Based on the reviews, we are likely to accept this manuscript for publication, providing that you modify the manuscript according to the review recommendations. 

All reviewers expressed that the paper was well writing and would be a valuable contribution to the literature on rabies. There are some minor comments that need to be addressed. In addition, the questionnaire was not available in the supplemental information provided and would need to be available upon revision.

Sincerely,

Amy J Davis, Ph.D.

Academic Editor

Dileepa Ediriweera

Section Editor

I have received three reviews of this manuscript. All reviewers expressed that the paper was well writing and would be a valuable contribution to the literature on rabies. There are some minor comments that need to be addressed. In addition, the questionnaire was not available in the supplemental information provided and would need to be available upon revision.

Reviewer's Responses to Questions

**Key Review Criteria Required for Acceptance?**

**Methods**

-Are the objectives of the study clearly articulated with a clear testable hypothesis stated?

-Is the study design appropriate to address the stated objectives?

-Is the population clearly described and appropriate for the hypothesis being tested?

-Is the sample size sufficient to ensure adequate power to address the hypothesis being tested?

-Were correct statistical analysis used to support conclusions?

-Are there concerns about ethical or regulatory requirements being met?

Reviewer #1: The methods are clearly described with sufficient detail to be replicated by others. They provide the required information to design similar studies and to explain the approach that the authors used to obtain the results.

Reviewer #2: The study objectives are clearly articulated, and the rationale well explained. Clear that this KAP survey will fill a gap in knowledge in this area and it is a well utilised and accepted survey tool in this context. Is there any evidence that KAP surveys elsewhere have translated into better rabies control programs? 

Population characteristics well described.

Methods: If the crush pens are used for dog vaccination program, there seems potential for bias here in that houses in close proximity may have different KAP to those far away. The survey data collection and analysis methods seems sound. 

No ethical concerns.

Reviewer #3: -Are the objectives of the study clearly articulated with a clear testable hypothesis stated? Yes

-Is the study design appropriate to address the stated objectives? Yes

-Is the population clearly described and appropriate for the hypothesis being tested? Yes

-Is the sample size sufficient to ensure adequate power to address the hypothesis being tested? Yes

-Were correct statistical analysis used to support conclusions? Yes

-Are there concerns about ethical or regulatory requirements being met? No

The methods are clearly articulated and appropriate for this study. However I could not find the questionnaire which is said to be available in the supplementary materials.

**Results**

-Does the analysis presented match the analysis plan?

-Are the results clearly and completely presented?

-Are the figures (Tables, Images) of sufficient quality for clarity?

Reviewer #1: The results were clearly presented in a logical manner that was easy to follow. The tables, figures and maps were well presented and clear.

Reviewer #2: Demographics and geographical area clearly presented. (No units for age in Fig 2). The urban vs rural results are interesting. 

Some of the graphs are hard to read because of the small text and questions being described in titles, but key results are pulled out in text. Fig 7- are Kavango east and Zambezi different in any way to other regions? urban/rural?

Reviewer #3: -Does the analysis presented match the analysis plan? YEs

-Are the results clearly and completely presented? Yes

-Are the figures (Tables, Images) of sufficient quality for clarity? Yes

**Conclusions**

-Are the conclusions supported by the data presented?

-Are the limitations of analysis clearly described?

-Do the authors discuss how these data can be helpful to advance our understanding of the topic under study?

-Is public health relevance addressed?

Reviewer #1: The conclusions adequately addressed the results, while also considering any potential shortcomings related to the study design and methodology. Discrepancies and results of particular interest were highlighted and explained further with additional considerations, making the conclusions clear and well-articulated.

Reviewer #2: There is a strong argument for understanding the dog demographics leading to a better planned dog vaccination program. However, I don't think the detail of how to get the vaccine coverage rate higher has really been addressed- for those that didn't vaccinate their dogs, does the survey give any insights into why? Is this access or attitudes mainly? This is touched on in lines 505/506 but I think needs to be expanded. 

The bite behaviour is interesting and shows a high take up of PEP, which is encouraging.

The conclusion draws together some suggestions how the authors think these results can specifically be used to improve the vaccination program and increase coverage rates, to help eliminate rabies.

Reviewer #3: The conclusions and limitations are completely described. The authors make sufficient mention of the wider application of the findings.

**Editorial and Data Presentation Modifications?**

Reviewer #1: Please note that the questionnaire was not available in the supplementary materials (at least to the reviewer). Please be sure to include this in the final manuscript as this would be something of importance to others looking to replicate the study elsewhere. 

Throughout the manuscript the authors switch terminology between “dog” and “canine”. Please standardize for clarity. 

Please check the punctuation after abbreviations throughout the manuscript, for example when using “e.g.” please include a comma after the abbreviation. “e.g.,” (line 137 and elsewhere).

Additional minor comments below should be considered for revision. 

Line 61: replace “on” with “in”

Line 71: “need” should be “needs”

Line 89: Please either change the “but” to "and" or say "vast majority is free-roaming, not only supporting disease transmission but further complicating control efforts."

Line 104: delete “organization” – it should be referred as the tripartite only.

Line 151: spelling: integrated. 

Line 164: double space between last word and reference.

Line 199: Spelling: Respondent

Line 231: “according to…” – the word “to” is missing. 

Line 231: Spelling: Agency

Line 276: Figure 1 caption “… June 2021 is shown” – the word “is” is missing. 

Line 284: The range of people living in a household was 1 – 70 people. Is this correct? Was one household found to have 70 people living there? 

Line 284: Please define the age of a “child” – is this anyone below 18 years of age?

Line 294: Spelling: DOHHs

Line 301: HDR mentioned is 1:5.45 – please remove the “1:” as this implies one person for every 5.45 dogs. 

Line 324: “Oshikato are supposed to have dog densities higher than 30 dogs/km²” – this statement is unclear. Why are they “supposed” to have those densities? Why are the areas with far higher densities not mentioned? 

Line 380: Figure 6 caption: “Answers of respondents when ask what would they do if they” – consider revising to "when asked what they would do…"

Line 380: Figure 6 caption: “(A) would encounter” - consider changing to “encountered”

Line 410: “behavior in case of bite exposure” – consider revising to “behavior in the case of bite exposure” (include the word “the”).

Line 450 – 451: “despite a national rabies control strategy in place” – a verb is missing. Consider revising to “despite a national rabies control strategy already being in place…”

Line 518: Change “rabies” to “rabid”

Supporting information, Table S2 description: “the” is repeated accidentally. 

S1 Figure description: “constituency wise” should change to “constituency wide” Alternatively, a preferred modification would be "showing the estimated dog population per constituency based…"

S2 Figure description: “constituency wise” should change to “constituency wide” Alternatively, a preferred modification would be "showing the estimated dog density per constituency."

Line 670 (references): Double punctuation after “2015”.

Line 854: The first author name is represented incorrectly. Change to “De Balogh KK,”

Reviewer #2: The author summary reads much like a technical summary rather than a non-technical summary and seems too repetitive of content within the abstract. I would suggest rewriting this with the 'non-technical' brief in mind and using plainer language. Eg omit One Health, "Our results show that there was 1 dog for about every 5 humans". etcLine 151- typo "intergated" should be integrated.

Inconsistent decimal points for proportions have been presented, sometimes it is 1 and sometimes 2.

Reviewer #3: Abstract typos/minor modifications: 

"will inform interventions on..." - should read will inform interventions in..."

"revealed deficiencies in some of the population." - should be reworded for clarify, perhaps "revealed deficiencies in knowledge, in some of the population."

Main text typos/modifications: 

Line 104: (WHO, WOAH, and FAO) need to be defined 

Line 230: "calculated projected" are both of these words necessary? 

Line 439-440: "significant time efforts" - needs to be reworded for clarity , suggest "significant time and effort" ??

Line 440: (n=18) Is this the number of teams, individuals in teams or days working?

**Summary and General Comments**

Reviewer #1: The study provided a detailed description and the results from a combined KAP survey and dog population estimate study, reaching an impressively large number of households in the at-risk areas for rabies in Namibia. Results indicated an inferior level of knowledge and attitudes towards rabies, with a need to also improve practices further. The dog population was estimated using the HDR with a value of 5.45 resulting in a total estimated population of approximately ~240 000 - 270 000 dogs in the NCAs. 

The study was well conceived and well-presented and provides important information to the elimination efforts in Namibia. It also provides good insights and information for others to replicate the study in other rabies-endemic regions around the world.

Reviewer #2: This paper is well written and gives a comprehensive account of this important study in Namibia. The insights gained will help improve prevention measures if implemented. This clearly has high significance for Namibia, but I am less clear on the originality of the paper and the usefulness of the findings globally. However, other communities can learn from these findings and some of the themes and suggestions for program improvements are clearly transferable.

Reviewer #3: Overall, this is a well written manuscript with important results applicable for rabies control.

PLOS authors have the option to publish the peer review history of their article (what does this mean?). If published, this will include your full peer review and any attached files.

Reviewer #1: Yes: Terence Scott

Reviewer #2: No

Reviewer #3: No

Figure Files:

Data Requirements:

Reproducibility:

References

---

## [Editor Report · Decision Letter 1]

24 Jan 2024

Dear Dr. Freuling,

We are pleased to inform you that your manuscript 'Dog Ecology and Rabies Knowledge, Attitude and Practice (KAP) in the Northern Communal Areas of Namibia' has been provisionally accepted for publication in PLOS Neglected Tropical Diseases.

Best regards,

Amy J Davis, Ph.D.

Academic Editor

Dileepa Ediriweera

Section Editor

The authors have done a nice job addressing reviewer comments. I have not additional concerns.

---

## [Editor Report · Acceptance letter]

30 Jan 2024

Dear Dr. Freuling,

We are delighted to inform you that your manuscript, "Dog Ecology and Rabies Knowledge, Attitude and Practice (KAP) in the Northern Communal Areas of Namibia," has been formally accepted for publication in PLOS Neglected Tropical Diseases.

Best regards,

Shaden Kamhawi

co-Editor-in-Chief

Paul Brindley

co-Editor-in-Chief
